# COMPOSITIONAL INTERFACES
# FOR COMPOSITIONAL GENERALIZATION

## ABSTRACT

With recent work such as GATO (Reed et al., 2022), we see the development of agents that can accomplish a variety of tasks, and are able to perceive the world and act in multiple observation and action spaces. We would want such agents to exhibit compositional generalization to unseen combinations of observation and action spaces, and adapt quickly to novel observation spaces by transfering knowledge. In this work, we demonstrate how these abilities can be achieved through the use of end-to-end modular architectures: the encoding of observations and the prediction of actions are handled by differentiable modules specialized to that space, with a single shared controller between them. To study the properties of such modular architectures in a controlled manner, we construct an environment with compositional structure, where each instance of the environment is created by combining an observation, action, and instruction space from a large set of options. We demonstrate that through the use of modularity, agents can generalize to unseen combinations of observation, action and instruction spaces; even when the unseen combinations are more challenging. Moreover, we demonstrate that modularity enables quick integration of novel observation modalities, requiring only adaptation of the modules encoding the new observation.

## 1 INTRODUCTION

In recent years, there has been remarkable successes with scaling model and data sizes. Across a wide variety of domains, the state-of-the-art is dominated by large models (pre-)trained on billions of samples (Brown et al., 2020; Goyal et al., 2021; Bommasani et al., 2021). This also holds true in the setting of multi-domain "generalist" agents, that can integrate perceptual information across multiple modalities, and can accomplish a variety of tasks (Reed et al., 2022; Shridhar et al., 2023).

In this multi-domain setting, transferring common knowledge between domains while respecting the particularities of each domain is still not a solved problem. For example, in the setting of agents that are either virtually or physically embodied, one of the most important special cases is simulation-to-real transfer. Practitioners would like to use gradient-based end-to-end learned controllers, but it is difficult to collect large amounts of training data on a physical robot in the real world. While there has been great progress in some tasks (Wijmans et al., 2019), driven in large part by ever higher-fidelity simulations (Savva et al., 2019; Shen et al., 2021), there are still no out-of-the-box solutions for generic sim-to-real transfer[1]. More generally, one would like to be able to train agents as much as possible in domains where training is cheap, and deploy after minimal training in domains where training is expensive. Even more generally, between domain transfer may allow sample efficiency via composition and abstraction. For example, a wheeled robot, a legged robot, and a digital assistant "embodied" in AR glasses all might be able to share some knowledge about indoor navigation despite differences in their locomotion. However, it is not possible to entirely abstract away all these differences- the wheeled robot cannot traverse the same terrain as the human with AR glasses.

Based on the above cited results in large-scale sequence modeling, one might wonder if researchers need to worry explicitly about transfer; maybe it is better to just scale token-based monolithic models like (Reed et al., 2022), or scale language models and some task/domain-specific models, with

---

[1]see sim for lively debates on this issue.

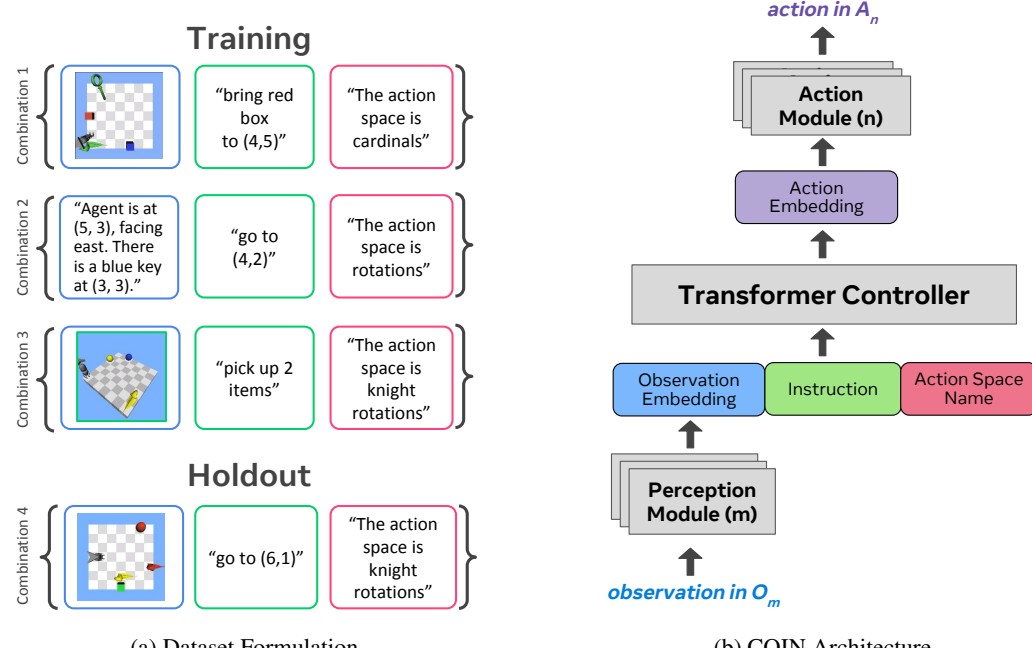

(a) Dataset Formulation    (b) COIN Architecture

Figure 1: Illustration of the dataset formulation and the COIN (**Co**mpositional **In**terfaces) architecture for compositional generalization. **(a)** Each environment instance is defined by a tuple $(O_m, A_n, I_k)$: a combination of observation $O_m$, action $A_n$ and instruction $I_k$ spaces. The agent is trained on data from a subset of all possible combinations, with the expectation of generalizing to combinations not included in the training dataset. **(b)** The agent architecture consists of: perception modules (one for each observation space), action modules (one for each action space) and a controller (shared across all environment instances). The controller takes the observation embedding, instruction and action space identifier as input, while outputting action embedding. When acting in an environment consisting of observation space $O_m$ and action space $A_n$, $m$-th perception module is used to predict the create the observation embedding and $n$-th action module is used to predict action from the action embedding.

inter-model text interfaces as in (Ahn et al., 2022; Zeng et al., 2022). We are sympathetic to these viewpoints, but as much as the bitter lesson has been that scale can be more important than good inductive bias, it has *also* been that optimizing end-to-end leads to the best results. Properly designed modular architectures can be both scalable, and allow end-to-end training (Pfeiffer et al., 2023). Furthermore, one of the benefits of modern attention-based architectures is that they are conducive to modular inductive biases without radical changes (Alayrac et al., 2022; Jaegle et al., 2021; Shridhar et al., 2023), and can differentiably interface various domains and still directly take advantage of pre-trained language models. Thus, we might hope to both encourage transfer through abstraction and composition, and allow end-to-end fine-tuning to handle the necessary details that cannot be abstracted; without giving up any of the benefits of large-scale pre-training.

In this work, we study the effectiveness of such a modular architecture for compositional generalization and transfer learning in the embodied agent setting. We develop an environment that allows us to independently vary perceptual modalities and action and task specifications, and use it to carefully analyze the agent's performance in these compositions. We show that we can compose the agent's perceptual suite, its task specifications, and its action spaces. Our experiments demonstrate zero-shot performance on held-out combinations of perception/instruction/action-space, and demonstration of fast adaptation (requiring less samples) to new perceptual or action spaces (with or without freezing the controller), and without excessive negative transfer.

## 2 SETTING

Our goal is to solve tasks defined by an environment instance $(O_m, A_n, I_k)$, which is constructed by combining $m$-th observation $(O)$, $n$-th action $(A)$ and $k$-th instruction $(I)$ space. Given an

observation $o^{(m)}$ from space $O_m$, action space id $n$ for $A_n$, and instruction $i^{(k)}$ from space $I_k$, the goal is to find a policy that will predict an optimal action $\pi(o^{(m)}, n, i^{(k)}) \rightarrow a \in A_n$. The agent is trained using imitation learning (Schaal, 1999) on a dataset of expert trajectories $\{D_{m,n,k}\}$ collected on $(O_m, A_n, I_k)$. We make sure that during training, the agent will be trained on samples from environment instances containing at least one of each of the individual spaces $O_m$, $A_n$, and $I_k$, but not *all possible combinations* $(O_m, A_n, I_k)$. This allows us to test compositional generalization by deploying the agent in environments containing unseen combinations, as demonstrated in Figure 1 (a). Alternatively, we can measure how quickly it adapts to newly added space.

Note that in this setting, generalization can mean two different things: (1) in-domain generalization where the agent is trained on trajectories from $(O_m, A_n, I_k)$, but a particular test sample $(o^{(m)}, n, i^{(k)})$ is never seen due to random procedural generation of environments. And (2), compositional generalization where test samples are from environment combinations that never seen during training. For example, learning to predict the right action in action space $A_{n'}$ given observation $o^{(m')}$ and instruction $i^{(k')}$ when the training dataset does not contain samples from environment $(O_{m'}, A_{n'}, I_{k'})$. In this work, we are particularly interested in the compositional generalization to unseen combinations of spaces.

## 2.1 Environment with Composable Observation, Action and Instruction Spaces

To study compositional generalization to unseen combinations of spaces, we construct a grid-world environment that supports multiple interfaces for observation and actions. The state is a 7x7 grid containing up to four different objects in addition to the agent itself. The objects can be picked up by the agent given that they are next to each other . The agent's inventory show object that are picked up, which later can be dropped down. Each object has a shape (box, ball, snake, key) and a color (red, green, yellow, blue). Each environment combination is constructed by selection on observation, action, and instruction space from one of the available options.

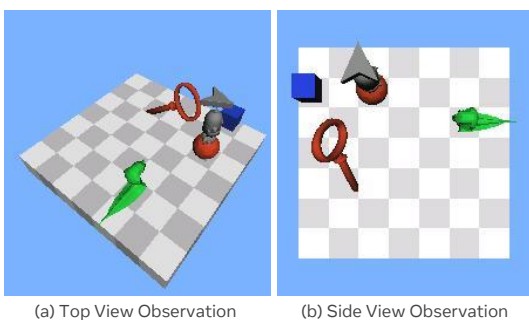

(a) Top View Observation     (b) Side View Observation

Figure 2: Illustration of the environment as represented by observation in *Top View* and *Side View* space. The agent (gray) is tasked with completing the instruction by navigating and interacting with objects in the grid. Each object is defined by a shape (ball, box, key, snake) and color (red, blue, green, yellow).

**Observation spaces ($O_m$):** There are six possible observation spaces, in which positions, shapes, and colors of the objects and agent in the grid are represented by: Text, Symbols, List, Grid, Top View, or Side View. These spaces are detailed in Table 1. Text space describes everything in human understandable language. Symbol space is similar, but uses compact symbols instead of words. To build an observation in List space, we represent everything with one-hot representation first. Then, for each object, we concatenate all its properties into a single vector. Finally, we stack all such vectors from all object and the agent together to give complete description of the state. Grid space also builds a vector for each object first, but then arranges them by their location instead, producing a 3D tensor. The remaining Top and Side view spaces are simply image rendering of the environment. These image spaces and Grid space assumes spatial location, which does not apply to inventory objects. As a workaround, we use List representation of inventory for those spaces. We made sure each observation contains sufficient information for completing the instruction, hence the tasks are fully observable.

**Instruction spaces ($I_k$):** In a given environment instance, the agent is tasked with completing an instruction from one of the eight possible instruction spaces. The simplest instruction, "Go to (x,y)", requires the agent to reach the specified location, while more complex instructions like "Pick up in order: red box, yellow snake, and green box" involves multiple steps and require the agent to distinguish shapes and colors. For the full list of instructions, please refer to the Appendix A.1. All instruction spaces involve manipulating objects and positions of the agent, with individual instructions being randomly sampled from the instruction space, while satisfying the constraint of instruction completion being possible given the initial state.

| Observation Space | Description |
|---|---|
| **Text** | A natural language description, e.g., "The agent is at (3, 5), facing east. There is a yellow snake at (2, 0). The agent has following items in the inventory: a blue box." |
| **Symbol** | A sequence of symbols, e.g., "A @ E x3y5. y S @ x2y0. I: b B." |
| **List** | A 2D tensor `o`, where `o[i] = concat([object_i_position, one_hot(object_i_shape), one_hot(object_i_color)])` |
| **Grid** | A 3D tensor `o` where the object at position $(x, y)$ is indicated by `o[x, y] = concat([one_hot(object_shape) one_hot(object_color)])` |
| **Top View** | An image made by projecting 3D space from the top (see Figure 2 right). |
| **Side View** | An image made by projecting 3D space from the side (see Figure 2 left). |

Table 1: Observation Spaces

| Action Space | Description |
|---|---|
| **Cardinals** | Move one step in one of the 4 cardinal directions (north, east, south, west) |
| **Move NW** | Move one step north, move one step west, or teleport to the south-east corner of the grid |
| **Rotations** | Rotate left, rotate right, or move one step forward in the direction of facing |
| **Teleport Direction** | Rotate left, rotate right, or teleport to a certain distance from the wall currently facing (0-6 steps from the wall) |
| **Knight Rotations** | Rotate left, rotate right, knight move left, or knight move right (i.e. two steps forward in the direction of facing + one step left or right). |

Table 2: Action Spaces

**Action spaces ($A_n$):** Completing each of the instructions can be accomplished by using one of the five possible action spaces. The type of movement available varies between spaces, as described in Table 2. Additionally, each action space has three shared actions for picking and dropping objects, and indicating the episode is done (Pick, Drop, and Done actions respectively). Successful completion requires the agent to complete the instruction and then output Done action.

## 3 ARCHITECTURE WITH COMPOSITIONAL INTERFACES

In our work, we use COIN (**Co**mpositional **In**terfaces) architecture. It is a modular architecture consisting of three main components: the perception modules, the controller, and the action modules, as demonstrated in Figure 1 (b), and detailed in the Appendix A.2. There is a different perception module for each observation space and a different action module for each action space. The controller is shared between all spaces and has a transformer architecture (Vaswani et al., 2017) (although any architecture that can handle variable-length inputs and tokens can be used ). Since instructions and action descriptions (we use textual descriptions, such as e.g. "The action space is cardinals.") are expressed in text, we can directly feed to the controller via simple word embedding layers.

The perception modules take in an observation and output a fixed-size embedding. The architecture for each perception module is chosen to best fit the modality of the corresponding observation space (the inventory, when represented as a list, is embedded with a 2D convolutional network). However, the number of vectors output by those specialized architectures vary from space to space (or even sample to sample for List space). In order to unify these outputs as input to the controller, we use adapter networks for each observation space. An adapter takes input of embedding of variable length and outputs embedding of fixed length, which is then input to the controller. That is achieved by using the network as cross-attention layers in enc-dec transformers.

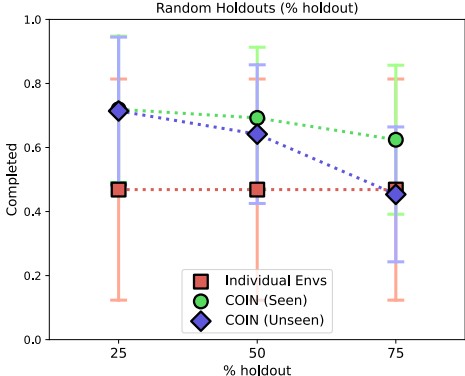 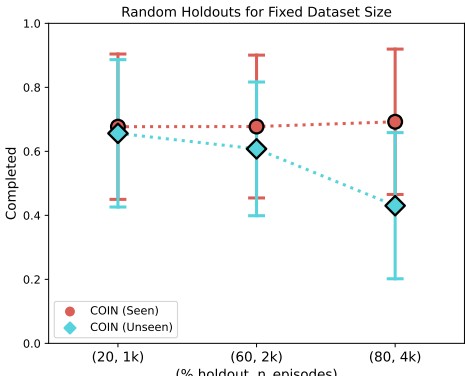

Figure 3: Agent performance at different percentages of held-out environments. Green: COIN agent on environments in the training data. Blue: COIN agent on environments *not* in the training data. Red: agent trained on only one environment.

Figure 4: Agent performance at different percentages of held-out environments with a fixed size of training dataset. Red: COIN agent on environments in the training data. Cyan: COIN agent on environments *not* in the training data.

The observation embedding is then concatenated with instruction and action space description embeddings. We also concatenate a fixed number of special padding tokens before feeding it into the controller. Among output vectors from the controller, we select the ones that correspond to the padding tokens, which gives us a fixed number of embeddings vectors to work with. Those embeddings are then fed into the action space specific action module, whose output corresponds to the dimensions of the corresponding action space. The fixed size of action embedding enables faster adaptation to new action spaces.

## 4 EXPERIMENTS

In the following set of experiments, we examine the compositional generalization properties of COIN agent. First, we examine the ability of COIN to generalize to unseen environment instances $(O_m, A_n, I_k)$, where combinations seen during training are selected uniformly at random. Next, we examine the case where the samples held out from training are a selected group of particularly challenging instruction and observation spaces. Lastly, we test the ability of COIN to adapt to new, completely unseen observation spaces $O_{new}$ through finetuning.

All the experiments use the compositional environments described in Section 2.1. For training, we use a dataset of 2,048 episodes with near-optimal trajectories $\{\tau_{(m,n,k)}^{(t)}\}_{t=1}^{2048}$ for each of the $240 = 6 \times 5 \times 8$ possible combinations of observation, action and instruction spaces (6, 5 and 8 options respectively); some of which will be held out from training. The near-optimal trajectories are generated using the A* algorithm or hand-engineered optimal policy. We evaluate the performance of the trained agent by measuring the rate of successful completion on both unseen and seen environment combinations. The task is considered completed if the agent reaches the goal defined by the instruction within the first 100 steps. When evaluating the trained agent on seen environment combinations, we measure the performance on that environment instance generated using a different random seed, i.e. the exact initial state and instruction are likely to differ from train time.

As an architecture for the perception modules, we use a pre-trained ResNet-18 network (He et al., 2015) for image spaces; for Grid and List spaces we use a 2D and 1D convolutional networks; for Text and Symbol spaces, we use 1D convolutions. The controller is a pre-trained Distilled-GPT-2 (Sanh et al., 2020), while each action module is a simple feed-forward network with the output corresponding to the dimensionality of the action space. The dimensions of observation embedding are $10 \times 768$, and the dimensions of action embedding are $4 \times 768$, where 768 is the dimension of GPT-2 token embedding. Each network is trained for 80 epochs. More details about the architecture and the training procedure can be found in the Appendix A.2.

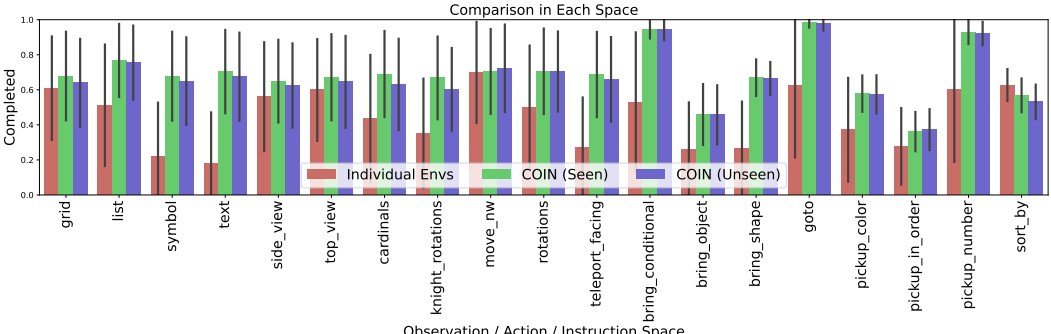

Figure 5: Comparison of performance between individual observation, action and instruction spaces. For each space, we report the performance averaged over all environment combinations containing that space (the error bars represent standard deviation). For  trained on samples from 75% environment combinations, we report the completion rate on environment instances included (green) and *not* included (blue) in the training data. The performance of an agent trained on only one environment instance is shown in red.

As a baseline, we use an agent trained on individual environment combinations using the same architecture, i.e. we train a separate agent on each of the 240 environment combinations. Note that in these cases, there is no weight sharing and each of such networks will contain only one perceptual and action module.

## 4.1 RANDOM HOLDOUTS

We start by examining the case where the environment instances $(O_m, A_n, I_k)$ included in the dataset have been chosen randomly by chance. To ensure relatively uniform coverage of all spaces, the procedure for selecting environment instances guarantees that each space individually has been included in least four combinations. We vary the percentage of environment instances held out from the training: either 25, 50 or 75% of all possible combinations. We report separately the performance on the environments included in the training data (seen) or held out for training (unseen).

In Figure 5 we take a look at the performance difference for each of the spaces individually, i.e. we fix one of the spaces (observation, action or instruction) and average over the rest. We consider the case where 25% combinations are held out (results with 50 and 75% held out combinations can be found in the Appendix A.3.1. Here we can see that COIN outperforms or matches the individual agents in all but one space (instruction space *Sort by Property*). We can also see that performance on unseen combinations matches the performance on seen combinations, which implies that the agent achieved near-perfect generalization to unseen compositions (with the remaining generalization gap being a consequence of either optimization difficulties or poor generalization to unseen observations and instructions). The greatest performance gains are seen on token observation spaces (*Text*, *Symbol*), which are particularly challenging for optimization and may particularly benefit from additional supervision provided by co-training on multiple observation spaces: the learning may be bootstrapped by learning a good controller on other, easier spaces.

The completion rates averaged over all the environment instances can be seen in Figure 3. From there, we can see that the COIN agent generalizes to unseen environment combinations extremely well, even outperforming the agents trained on individual environment combinations when the holdout rate is over 50%. The performance of COIN agent drops as we decrease the number of combinations included in the training data as expected. The error bars represent standard deviation over 240 environment instances.

To evaluate the relative importance of using more data for each of the training environment combinations versus using more environment combinations in the training dataset, we run an experiment where the total number of episodes seen during training is kept constant while varying the holdout rate and number of episodes used in training. The total number of episodes used in training is always 192k, with the percentage of environment combinations used in training and number of episodes being: $(80\%, 2^{10})$, $(40\%, 2^{11})$ and $(20\%, 2^{12})$. The results can be seen in Figure 4. We find that for compositional generalization, it is more advantageous to use more environment combinations.

## 4.2 Hard Holdouts

Next, we consider the hard case where the hold-out set is composed of particularly challenging combinations, either in terms of data collection or training time. We are particularly interested in this case, as in practice, there may be cases where data collection is much more challenging for some combinations (e.g. when some observation spaces correspond to data collected on real robots instead of data collected in simulation, where it can be hard to evaluate completion of some instructions outside of simulation) . In these cases, it might be advantageous to collect the data on easier combinations for training and obtain good zero-shot performance on hard combinations without requiring data collection or training.

To construct hard combinations, we selected image observation spaces ($\mathcal{O}_{hard}$ = { *Top View*, *Side View* }) and two of the hardest instruction spaces ($\mathcal{I}_{hard}$ = { *Bring Object, Pickup In Order* }) as the performance on these spaces is

| Method | Completion Rate |
|---|---|
| Individual Envs | $0.35 \pm 0.18$ |
| Random Holdouts (25%) | $0.34 \pm 0.07$ |
| Hard Holdouts (8%) | $0.26 \pm 0.08$ |
| Hard Holdouts (32%) | $0.21 \pm 0.12$ |
| Hard Holdouts (55%) | $0.20 \pm 0.10$ |

Figure 6: Agent performance on a set of 20 particularly challenging environment instances $\mathcal{E}_{hard}$. For COIN with both random and hard holdouts, we report zero-shot performance. In hard holdouts, the entire $\mathcal{E}_{hard}$ was held out from the training data; whereas in random holdouts, a random selection of 25% of combinations was held out. For hard holdouts, we report results where a total of 8, 32 and 55% of environment instances (including $\mathcal{E}_{hard}$), were held out. We also report the performance of agents trained on individual environments from $\mathcal{E}_{hard}$.

generally the lowest, training trajectories are the longest and training on images takes more time. The hold-out set $\mathcal{E}_{hard}$ then consists of all combinations $(O_m, A_n, I_k)$ where $O_m \in \mathcal{O}_{hard}$ and $I_k \in \mathcal{I}_{hard}$. In our case, this will be a total of 20 environment combinations. We train the COIN agent on the remaining 200 combinations, or a randomly sampled set of 75 and 50% of the remaining environments (in total, this corresponds to 8, 32 and 55% of all possible combinations being held out respectively). For hard holdouts, we report results on 5 different random seeds.

As shown in Table 6, we find that while not matching the performance of agents trained individually on those combinations or when the combinations are held out randomly, we still observe good transfer from easier to hard combinations, despite never seeing the particularly hard combination of observation and instruction in the training dataset.

## 4.3 New Perception Spaces

Lastly, we examine if COIN agent can effectively and efficiently incorporate new observation spaces. This is particularly relevant in a continual learning setting, where over a lifetime, new perceptual spaces may need to be added, without hurting the performance on spaces the agent has been already trained on or requiring training again from scratch on the entire dataset including the new observation spaces. Modular architectures have the potential to integrate new observational spaces without affecting the performance on other spaces by training only the new perceptual module, while freezing the controller and action modules. Moreover, training only the perception modules may require less data and converge more quickly.

To test this, for each observation space $O_{new}$, we first take out all the samples with that observation space from the training data (in total 40 different environment combination) and train on the data from randomly selected 75% of the environment combinations. Next, we take the trained COIN network and add the freshly initialized perception module for $O_{new}$, which is trained on the data from all the environment combinations containing $O_{new}$. The weights of the controller and action modules are kept constant. To compare the data requirements of adding the new perceptual spaces to already trained controller and action modules, to training the entire network from scratch, we train the new module using 2048, 1024, or 516 episodes from the dataset (full, half, or one-fourth of the dataset respectively). For comparison, we also try fine-tuning the entire network on the combinations with the new observation space (i.e. without freezing weights of the controller and action modules). We also compare the results to the modular network trained on the same 40 environments from scratch (i.e. without transfer from other observation spaces).

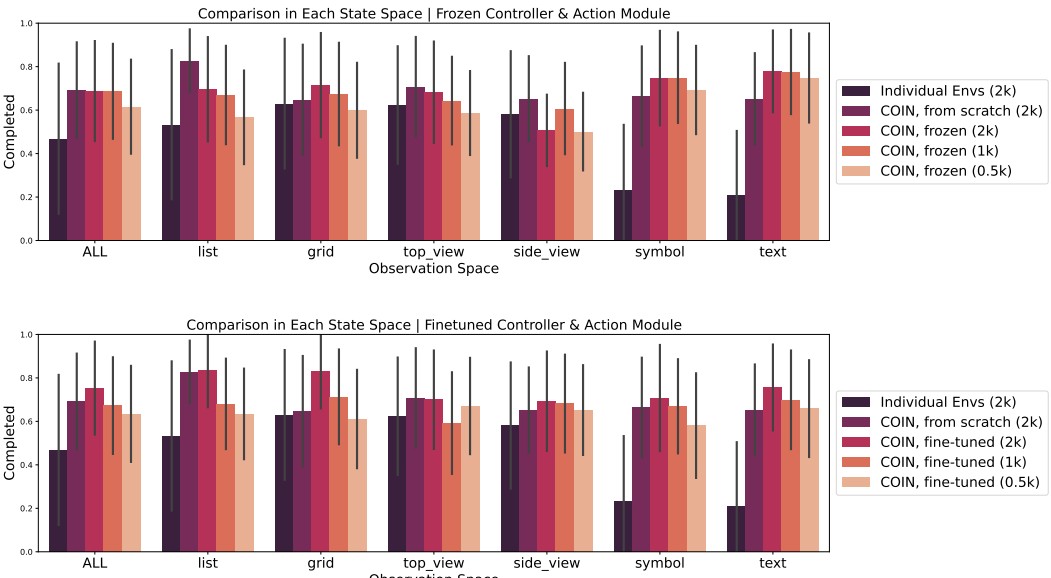

Figure 7: Performance of COIN agent on the 40 environment combinations $\mathcal{E}^O$ containing a newly added observation space $O$, for each of the six available observation spaces. The controller and action modules are trained on 75% of all randomly selected combinations *not* including $\mathcal{E}^O$. In the top figure, we only train the newly added perceptual module (i.e. without affecting the performance on other tasks), whereas in the bottom figure, we fine-tune the entire network. We report the results using 2048, 1024, or 516 episodes from each environment in $\mathcal{E}^O$ for training. We contrast these results to an agent trained from scratch on $\mathcal{E}^O$ and agents trained individually on each task in $\mathcal{E}^O$. The results are reported over 3 random seeds, with the error bar representing standard deviation over all environment instances in $\mathcal{E}^O$

Results on the new observation spaces with freezing of the controller and action modules can be found in Figure 7 Top and without freezing in Figure 7 Bottom. We find that, when averaged over the observation spaces, by training only the new perception module, we can match the performance obtained by training from scratch and outperform training on individual environments. Moreover, we can match the training from scratch with one-fourth of the data. This is likely due to transfer from other tasks, where the new observation just needs to be mapped to a representation already understandable by the controller. We are finding that finetuning the entire network is not necessary for achieving good performance, hence a new observation space can be incorporated without affecting performance on other environment instances.

## 5    RELATED WORK

**Single Modality:** Compositional generalization is often studied at the level of a single domain. In vision domain, models are tested if they can recognize an image that contains an unseen combination of different visual properties (e.g. shape, color), with emphasis on disentangled representation (Xu et al., 2022). Instruction and task is another domain where compositional generalization is well studied (Zhang et al., 2018; Zhou et al., 2022). At test time, the agent is given an unseen instruction or task that usually can be accomplished by chaining together already learned skills (Lake & Baroni, 2017). This idea of learning a set of skills that can be composed together can be traced back to options framework (Sutton et al., 1999) and other hierarchical RL methods (Sukhbaatar et al., 2018). Given the recent success of large language models, language is increasingly being used: Jiang et al. (2019) leverages natural language for hierarchical RL; Huang et al. (2022a) uses large language models to decompose tasks into smaller subtasks; Mezghani et al. (2023) has a single model for both policy and language reasoner. Unlike these, the focus our paper is composition generalization across of different modalities.

**Modular Architectures for Multi-task Learning:** Modular architectures can be viewed as composition of internal modules of the agent, and often studied together with multi-task generalization. PathNet (Fernando et al., 2017) uses genetic algorithm to select a subset of a neural network to be used for a specific task, showing positive transfer from one task to another. Rusu et al. (2016) grows a neural network by adding new modules with each new task, but allowing them to connect all previous modules. Continual learning is enabled by freezing previous modules, but still positive transfer is observed between different Atari games. Similarly, Gesmundo & Dean (2022) both grows and and selects subsets of the network. However those methods require training on the target environment, while our method enables zero-shot generalization to an unseen environment while utilizing a simple end-to-end training. LegoNN (Dalmia et al., 2022) is an encoder-decoder model with decoder modules that can be reused across machine translation and speech recognition tasks. Our approach for connecting perceptual modules to the controller recalls Alayrac et al. (2022), where the authors use cross-attention to connect a vision model to a text Transformer, and Jaegle et al. (2021) where this idea is discussed more generally.

**Multi-Embodiment Continuous Control:** Devin et al. (2017); Huang et al. (2020) used Graph Neural Networks (GNN) to build modular architecture that can control many different physical bodies. Furthermore Huang et al. (2020) shows such architectures are capable of zero-shot generalization to a new physical body. Our work, as Kurin et al. (2020), uses a Transformer in place of the GNN. The "action spaces" in this work are analogous to the body morphologies in those. However, here, we study composable generalization not just to different action spaces, but to perceptual and task spaces as well.

**Language Model as Controller and Planner via Text Interfaces:** Several works have shown how a language model used as can be a nexus between modalities, and controller or planner for embodied agents. The general theme is to use text as glue, and the language model as a central processor. For example Socratic agents (Zeng et al., 2022) combines multiple pre-trained models from different domains to create a system that can solve unseen task involving a novel combination of domains. Similarly Huang et al. (2022b) deploy a pretrained LM as a robotic controller by augmenting it with additional models that can interpret visual scenes in language. In Ahn et al. (2022), the language model is used to score affordances based on a task description, and as a planner, following Huang et al. (2022a). In this work, rather than connecting modules via text, we use self-attention, allowing end-to-end learning.

**Transformers in Behaviorally Cloned Generalist Agents:** Our work is closely related to Reed et al. (2022) and Shridhar et al. (2023), where the authors showed that end-to-end Transformers can be effective controllers for embodied agents with multi-modal perception and/or actions. As in those works, we train via behavioral cloning. While Reed et al. (2022) tokenizes all inputs and treats the Transformer controller as a monolith, we allow passing gradients to perceptual or task-specific submodules. In this, we are similar to Shridhar et al. (2023), but rather than consider a fixed perceptual and action space as in that work, we show that our setup allows compositional generalization between perceptual, action and task spaces, and fast adaptation to new spaces.

## 6 CONCLUSION

In this paper, we proposed a modular architecture with differentiable interfaces to various modalities of perception and action. These interfaces are connected to a shared controller, enabling passing gradients and end-to-end backpropagation, while supporting knowledge sharing. We developed a new environment in which perceptual modalities, sets of actions and types of instructions can be independently varied. This environment allowed us to systematically study compositional generalization across different modalities.

An agent trained with the modular architecture demonstrated zero-shot generalization when tested on unseen combination of modalities, outperforming an agent trained only on that combination. Furthermore, on a set of held-out combinations that were challenging to learn for a "single-environment" agent, the modular agent still showed zero-shot generalization. Lastly, we have shown that new perceptual modalities can be easily incorporated by training only the interface processing that modality. These results show that modular architectures can engender compositional generalization and cross-domain transfer without any special training scheme.

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

| Instruction Space | Description |
|---|---|
| **Go To** | The agent needs to move to a randomly sampled location. E.g., "Go to (1, 3)." |
| **Pickup N** | The agent needs to pick up exactly N objects, where N is a number randomly sampled between 1 and # objects in the environment. E.g., "Pickup 2 objects." |
| **Pickup Color** | Pick up one randomly chosen object specified by color. E.g., "Pick up one red item." |
| **Bring Shape** | Bring one randomly chosen object specified by shape to a randomly chosen target location. E.g., "Bring one key to (4, 4)." |
| **Bring Conditional** | Bring one item to one of the 4 corners, depending on the shape or color of the object. Whether the target corner is determined by shape or color is randomly sampled. E.g., "Bring one item to shape location." |
| **Bring Object** | Bring an item specified by a randomly chosen shape and color to a randomly chosen target location. E.g., "Bring green snake to (6, 2)." |
| **Pickup in Order** | Pick up items in a specified random order determined by the color and shape of the objects. E.g., "Pick up in order: red box, yellow snake, and green box." |
| **Sort by Property** | Move all items of the same color or shape within 1 step of each other. Whether the items should be moved based on shape or color is randomly sampled. E.g., "Sort items by color." |

Table 3: Instruction Spaces

Andy Zeng, Adrian S. Wong, Stefan Welker, Krzysztof Choromanski, Federico Tombari, Aveek Purohit, Michael S. Ryoo, Vikas Sindhwani, Johnny Lee, Vincent Vanhoucke, and Peter R. Florence. Socratic models: Composing zero-shot multimodal reasoning with language. *ArXiv*, abs/2204.00598, 2022.

Amy Zhang, Adam Lerer, Sainbayar Sukhbaatar, Rob Fergus, and Arthur D. Szlam. Composable planning with attributes. *ArXiv*, abs/1803.00512, 2018.

Allan Zhou, Vikash Kumar, Chelsea Finn, and Aravind Rajeswaran. Policy architectures for compositional generalization in control. *ArXiv*, abs/2203.05960, 2022.

# A APPENDIX

## A.1 ENVIRONMENT

In this section, we provide additional details about the construction and dynamics of the environment. The full list of instruction spaces is given in Table 3. When initiating the environment, we first sample the initial state: the number of objects is sampled uniformly from $[1, 4]$, positions of the objects and the agent are sampled uniformly without replacement. The properties of objects (color, shape) are also sampled uniformly without replacement. The instruction is sampled next, with the constraint that sampled instruction can be completed given the initial state.

We constrain the environment such that two objects can not be in the same position (e.g. using the Drop action on top of another object will have no effect), however, the agent can be in the same position as an object. In the Grid observation space, if the agent is in the same position $(x, y)$ as an object, we sum the two one-hot representations at $(x, y)$. The color of the agent is always grey.

The inventory has size 4, and the objects are added or removed from the inventory in the last-in, first-out order. Using the Done action in any state other than the goal state will have no effect. Using a movement action that would lead the agent outside the grid also has no effect.

Full implementation of the environment will be released in the camera-ready version of the paper.

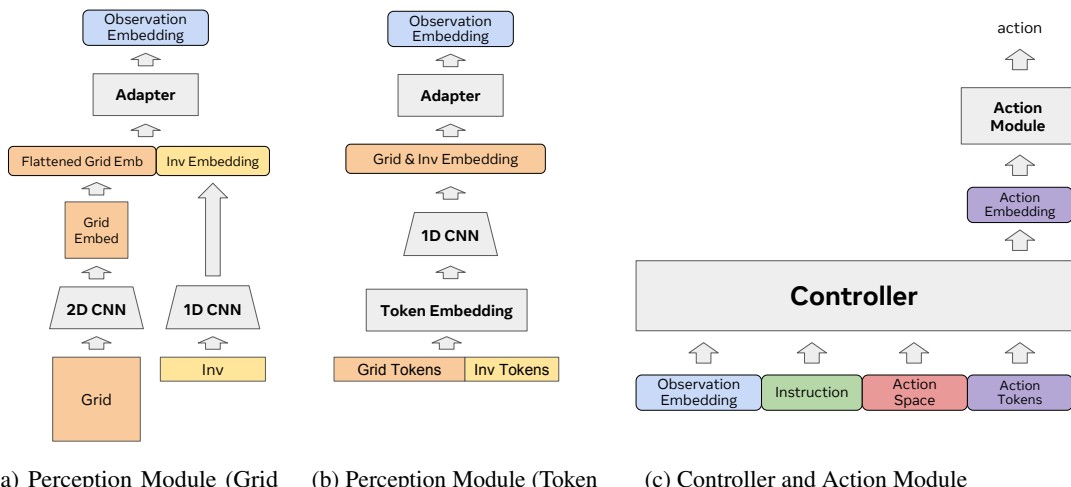

Figure 8: COIN Modular Architecture.

## A.2 ARCHITECTURE

An illustration of the architecture can be seen in Figure 8. For non-token observation spaces, the perception module is parametrized as in Figure 8 (a). For observation space Grid, the 2D convolutional network consists of layers of 2D convolutions followed by ReLu non-linearity. The dimensions of output channels in the final convolution correspond to the dimensions of the token embedding of the controller. For the observation space List, the 2D convolution in Figure 8 (a) is replaced by 1D convolution (same architecture as for inventory embedding: two layers of 1D convolution followed by ReLu non-linearity; with the dimensions of the output channels corresponding to the dimensions of the token embedding of the controller, which is 768), whereas for the image observation spaces Top View and Side View, the 2D convolution is replaced by a pre-trained ResNet-18 [2] network. For token observation spaces Text and Symbol, the grid and inventory observations are embedded together (see Figure 8 (b)): each token is first represented using the pre-trained token and positional embeddings, then processed using three layers of 1D convolutions followed by ReLu non-linearity (with kernel sizes 7, 2 and 1 respectively). The resulting output is then passed through a layernorm. We found this architecture to work the best with token observational spaces.

For each observation space $O$, the resulting inventory and grid embeddings are flattened and concatenated to a sequence $\mathbf{v}$ of length $L^{(O)}$ and passed through an adapter module, resulting in a sequence $\mathbf{h}$ of length $L$. The adapter module is a simple attention layer of the form:

$$\mathbf{h}_j = \sum_i^{L^{(0)}} \alpha_{i,j}^{(O)} (\mathbf{v}_i + \mathbf{b}_i^{(O)}) \tag{1}$$

$$\alpha_{i,j}^{(O)} = softmax(\frac{\mathbf{w}_j^{(O),T} \mathbf{v}_i}{\sum_{i'}^{L^{(0)}} \mathbf{w}_j^{(O),T} \mathbf{v}_{i'}}), \tag{2}$$

where $\mathbf{h}_j$ is the j-th element of list $\mathbf{h}$, $\mathbf{v}_i$ is i-th element of list $\mathbf{v}$ (each of dimension 768 corresponding to the dimensions of token embeddings), while $\mathbf{w}_j^{(O)}$ and $\mathbf{b}_i^{(O)}$ are the learned parameters of the adapter module of observation space $O$ (also each of dimension 768). In our experiments, the length of the final embedding is 10.

The resulting observation embedding $\mathbf{h}$ is then concatenated with token embeddings of instruction, action space description, and special action tokens (as shown in in Figure 8 (c)), while adding the positional embeddings, and passed through the controller network. The controller network is a pre-trained Distilled-GPT-2. The action embeddings are taken from the position of special action

---

[2]huggingface's `ResNetModel.from_pretrained("microsoft/resnet-18")`

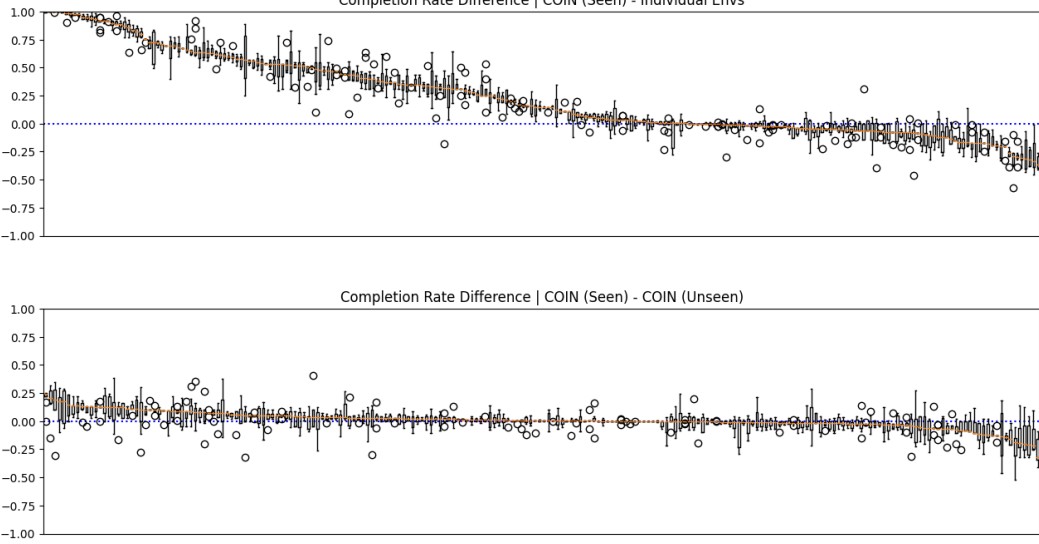

Figure 9: Distribution of differences in performance between COIN agent on seen tasks and agent trained on individual tasks (top), as well as distribution of differences in performance between COIN agent on seen and un-seen tasks. The number of random seeds for each environment combination varies as the held-out combinations are selected by chance.

tokens, flattened and then passed through the action module. The action module of each action space has two layers of linear projection followed by ReLu non-linearity, followed by a final linear layer (with the output dimension determined by the number of actions of the corresponding action space) and a softmax layer. In our experiments, the length of action embedding is 4.

Full implementation of the architecture will be released in the camera-ready version of the paper.

## A.3 EXPERIMENTS

In this section, we provide a more detailed description of the training procedure, as well as additional results. For a detailed description of the architecture, see the previous Section (A.2). In addition to the expansion of the results from Section 4, in A.3.3, we also add an experiment examining the ability of SOTA language models to handle compositional generalization within in-context learning.

Each model is trained for 80 epochs using AdamW optimizer and a linear learning rate schedule. The learning rate is $3 \times 10^{-4}$, with the batch size 8. We generally found that lower batch sizes result in better generalization. The completion rate of the trained agent was evaluated over 80 episodes for each environment combination (we consider the task completed if the goal state is reached in less than 100 steps), where the environment seeds used to generate training data are different from the environment seeds used in the evaluation.

The full code for running the experiments will be released in the camera-ready version of the paper.

### A.3.1 RANDOM HOLDOUTS

In Figure 9, we visualized the difference in performance for each of the 240 environments and for the case of training on 75% environment combinations. Positive difference can be interpreted as the first method outperforming the second in this environment. In Figure 9 (top), we can see that COIN agent significantly outperforms agents trained on individual combinations for a majority of environments, in some cases in fact, improving from no completed tasks to almost perfect task completion. In Figure 9 (bottom), we compare the performance of COIN agent on seen and unseen tasks. Here we can see that difference in performance is typically within the variance of performance on individual tasks, indicating very good generalization to unseen combinations.

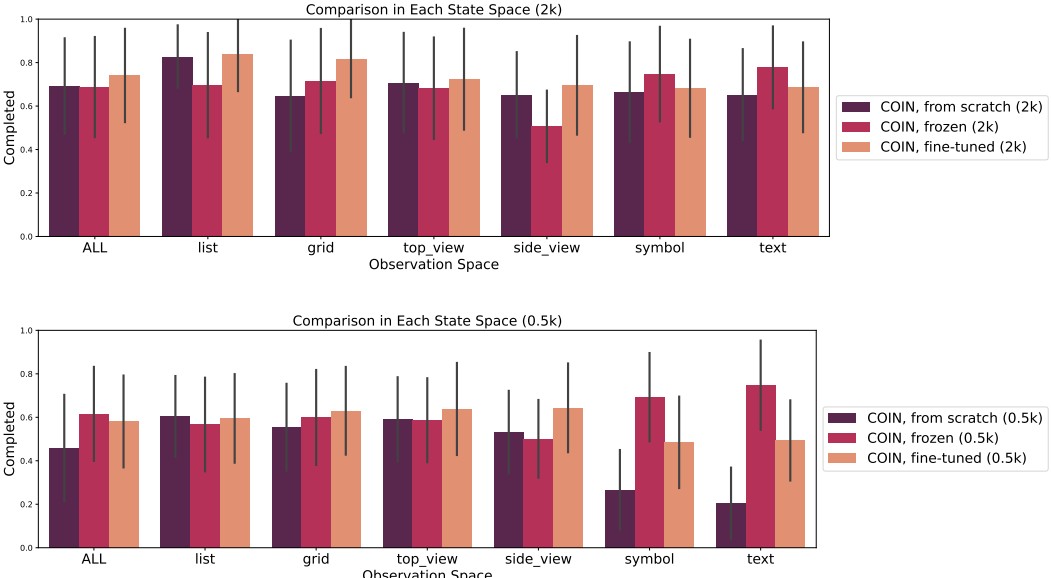

Figure 10: Performance of COIN agent on the 40 environment combinations $\mathcal{E}^O$ containing a newly added observation space $O$, for each of the six available observation spaces. The controller and action modules are trained on 75% of all randomly selected combinations *not* including $\mathcal{E}^O$. In the top figure, we train on all 2048 episodes from each environment in $\mathcal{E}^O$, whereas in the bottom figure, we train on only 516 episodes from each environment. The results are reported over 3 random seeds, with the error bar representing standard deviation over all environment instances in $\mathcal{E}^O$

Figure 11 corresponds to Figure 4 (in Section 4.1) for the case of training on 50% of environment combinations (top) and 25% of environment combinations (bottom). In Figure 3 (Section 4.1), we already demonstrated how compositional generalization overall becomes worse as the percentage of environment combinations used for training is decreased, whereas Figure 11 demonstrates which of the spaces are affected the most.

### A.3.2 New Observation Spaces

To compare the performance of COIN agent under different data sizes, we group the result presented in 4.3 based on the number of episodes used in training. Figure 10 groups the results of COIN agents (either trained from scratch on the new observation space, only training the new perceptual module or training the new perceptual module alongside fine-tuning of the controller and action modules) based on the number of episodes used in training. Figure 10 (top) shows the results where all 2048 episodes were used in training, and Figure 10 (bottom) shows the results where only 516 episodes were used in training. Each of the experiments is as described in Section 4.3. We can see here that transfer is particularly advantageous in the low-data regime and in optimizationally challenging observation spaces.

### A.3.3 In-context learning with GPT-3

Lastly, in order to illustrate that compositional generalization with respect to observation, action and instruction spaces remains a challenge even for state-of-the-art language models, we evaluate in-context learning of the best available GPT-3 model (`text-davinci-003`).

To make the task less challenging for the language model, we evaluate compositional generalization only with respect to action and instruction spaces and construct environment combinations by using: only Text observation space, a smaller set of action spaces (Cardinals, Rotations, Move NW, Knight Rotations) and easy task spaces (Go To, Pickup Number, Pickup Color, Bring Shape); for a total of 16 environment combinations. We randomly select 25% of environment combinations for evaluation in 24-shot setting. Each prompt is constructed by appending together a random selection of 24 samples from the training set, each of which is formulated as in the following example:

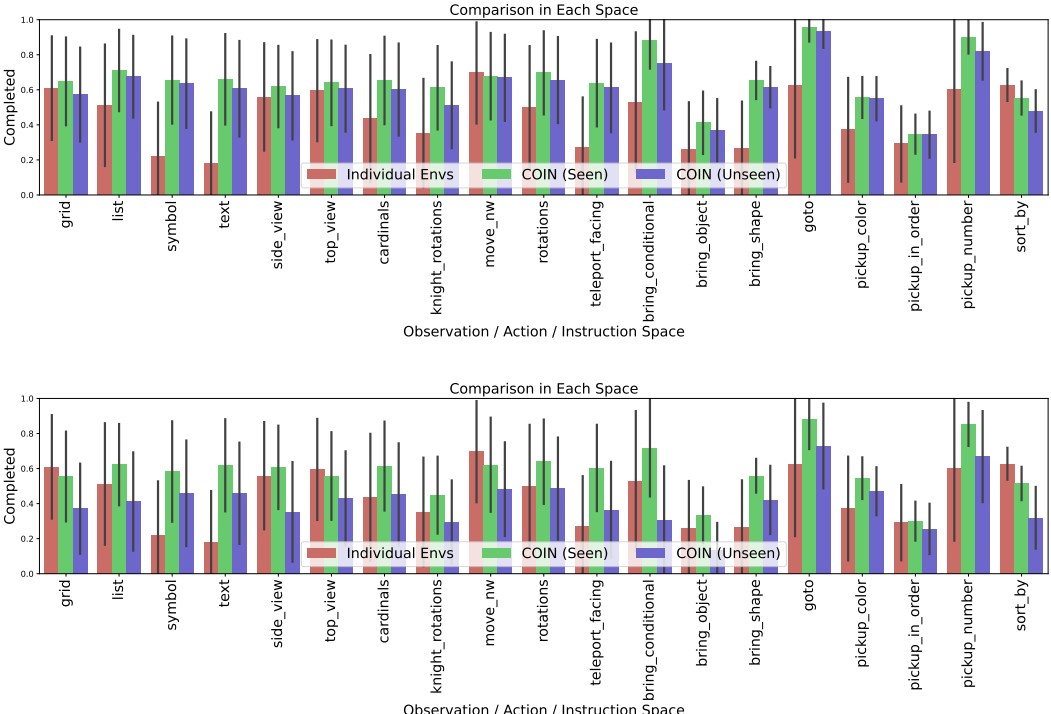

Figure 11: Comparison of performance between individual observation, action and instruction spaces. For each space, we report the performance averaged over all environment combinations containing that space (the error bars represent standard deviation). In the top figure, we trained on $50\%$ environment combinations, and in the bottom figure, we trained on $25\%$ environment combinations. We report the completion rate on environment instances included (green) and *not* included (blue) in the training data. The performance of an agent trained on only one environment instance is shown in red.

> "Q: The agent is at (1, 4), facing north. There is a green box at (4, 2), and a red snake at (6, 3). The agent has the following items in its inventory: yellow ball. The task is Pickup 1 item. Which of the following actions should you choose:
>
>    (a) Turn right,
>
>    (b) Turn left,
>
>    (c) Take one step forward,
>
>    (d) Pick the top item,
>
>    (e) Drop the top inventory item,
>
>    (f) Finish episode.
>
> A: (f) Finish episode."

We construct one such prompt for each observation while acting in the held-out environment, appending the current observation to the prompt and recording the model completion. We then use this model completion to predict the next action. As before, we evaluate the method based on the completion rates in each of the held-out environments (here using 20 rollouts), and report the results over 5 random selections of the hold-out set.

The completion rate for a model using GPT-3 for action prediction was $5\%$ (with standard deviation $6\%$). For comparison, the completion rates of COIN agent on the same set of environment combinations (when encountered in the random held-out set, from experiments in Section 4.1 with the hold-out rate $25\%$) is $79\%$ (with standard deviation $18\%$).

