# OpenReview forum: "Compositional Interfaces for Compositional Generalization"
_ICLR.cc/2024/Conference — Submitted to ICLR 2024_

### Official Review · Reviewer_B95a · 2023-10-12

**Soundness:** 3 good
**Presentation:** 3 good
**Contribution:** 2 fair
**Rating:** 3
**Confidence:** 3

**Summary:**

This paper targets on developing agents for compositional generalization to novel combinations of observation and action spaces by using COIN architecture.

**Strengths:**

1. The target of this paper on generalization (specifically compositional generalization) is a key area in machine learning.
2. A modular approach to compositional combinations of observation and action spaces seems a good fit and appropriate approach.
3. Experiments show improvement over various COIN baselines.

**Weaknesses:**

1. This paper seems to be solely an application of COIN architecture in the setting of compositional combinations of observation and action spaces. It is not clear what novelty or technical contribution it has.
2. The experiments are mainly on COIN architecture. No comparison with other approaches. It is hard to assess the effectiveness of this method.

**Questions:**

1. Why is there no comparison with non-COIN approaches?
2. What is the novelty of this paper? It seems to be simply an application of COIN architecture on compositional combinations of observation and action spaces.

---

> ### Author Response · Authors · 2023-11-21
>
> We thank the reviewer for the feedback, we are glad they found the problem relevant and our approach a good fit for the problem! In our response, we hope to clarify the key contributions of the paper as well as the choice of baselines.
>
> > This paper seems to be solely an application of COIN architecture in the setting of compositional combinations of observation and action spaces. It is not clear what novelty or technical contribution it has.
>
> The primary contribution of the paper is the study of compositional generalization of multi-task multi-modal agents, specifically with respect to observation, action and instruction spaces. To the best of our knowledge, our work is the only such study to date. It is _not merely an application of a modular architecture, but rather a systematic evaluation of how much a modular architecture improves compositional generalization_ for such agents. We provided detailed experiments and analysis with respect to the number of training combinations, data size, and hold-out set difficulty. This form of compositional generalization is an important property of generalist agents: many real-world tasks  will involve training agents with a variety of sensors and actuators to accomplish a wide variety of objectives.
>
> The remaining novel contributions of our work are as follows:
> - An environment that enables such study by: (a) allowing easy composition of arbitrary observation, action and instruction spaces, (b) supporting a large number of observation, action and instruction spaces, (c) providing an automated way to generate training data
> - An architecture that can handle learning in this setting through modular design, as well as incorporating new perception modalities without impacting performance on the tasks the agent was previously trained on.
>
>
> > The experiments are mainly on COIN architecture. No comparison with other approaches. It is hard to assess the effectiveness of this method.
>
> With respect to the choice of baselines, the literature has no standard off-the-shelf approaches applicable to our setting beyond our comparisons to single-task models.  The closest possible approaches are tokenized transformers such as those used in GATO (which is not open-sourced). With such model, however, "how to tokenize" would become a non-trivial question beyond the scope of our work, even if GATO was open-sourced.
>
> Hence the most fair comparison is with respect to agents trained in a single-task setting or the expert data (in our case, the experts always complete the task). Note that for the same reasons, GATO also does not compare their model to alternative multi-task multi-modal architectures.
>
> -----
>
> We hope our response clarified the novelty of our contribution and the choice of baselines. If there are any remaining concerns, we are happy to discuss them here.

---

> > ### Comment · Reviewer_B95a · 2023-11-22
> > **Thanks for the authors' response.**
> >
> > Based on the authors' response, my rating remains the same.

---

> > > ### Author Response · Authors · 2023-11-22
> > >
> > > Thank you for your response.
> > >
> > > Do you have any more specific objections, e.g. specific baselines you think we should have run or reasons you believe the contribution as described is not novel?

---

### Official Review · Reviewer_Pm6p · 2023-11-01

**Soundness:** 3 good
**Presentation:** 4 excellent
**Contribution:** 3 good
**Rating:** 6
**Confidence:** 3

**Summary:**

The paper proposes to use end-to-end modular architectures to achieve compositional generalization to unseen combinations of observation
and action spaces. It requires individual encoding and action modules for each observation and action space, with a single controller connecting them shared across different spaces. The paper constructs an synthetic environment with compositional structure and show through extensive experiments that the proposed method enables generalization to unseen combinations of observation and action spaces.

**Strengths:**

+ Originality:

The paper proposes an effective design for compositional generalization of the agent's observation and action spaces. While modular architectures are well known in previous works, the presented work is novel in a sense that it is a successful demonstration that modular design can be effectively used for compositional generalization.

+ Quality & Clarity:

The paper is well-written. Experiment designs are clean and comprehensive in general. The presentation is well-organized and easy to follow.

+ Significance:

The presented work can be potentially beneficial for generalist agent design.

**Weaknesses:**

- Significance:

It would be great if noisy real-world data or more complex visual data is included in the testing scenario. It is unclear for now whether this design is robust enough to deal with the not-that-clean data regime.

**Questions:**

i) Could the authors provide insights or discussions about why the proposed method performs worse in the instruction space of `Sort by Property`? Is there a class of tasks that the proposed method fail to handle?

ii) For the `HARD HOLDOUTS` set-up, will the performance further improve if there are more easy combinations available for training? Are there any other possible solutions to address the easy-to-hard transfer problem?

---

> ### Author Response · Authors · 2023-11-21
>
> We are thankful to the reviewer for the positive feedback, we are glad they found the paper well-written and recognized the novelty of our work! In the following, we hope to address the reviewer's concern about complexity of the environment and answer their questions.
>
> > It would be great if noisy real-world data or more complex visual data is included in the testing scenario. It is unclear for now whether this design is robust enough to deal with the not-that-clean data regime.
>
> While we agree that results on noisier environments would be interesting, there were no such environments available that would allow us to study compositional generalization wrt observation, action and instruction spaces specifically. Moreover, in order to draw strong conclusions, it is often more useful to study the phenomena in a controlled, synthetic environment. By focusing on a simpler environment design, we were able to:
> (a) generate a very large number of environment combinations and individual spaces,
> (b) have a much better control of the factors of variation – in our case, we can only vary the observation modality, action space and instruction space – without introducing other confounding sources of variation that would be present in more complex environments.
>
> ## Questions
>
> > i) Could the authors provide insights or discussions about why the proposed method performs worse in the instruction space of Sort by Property? Is there a class of tasks that the proposed method fail to handle?
>
> While we can not tell with certainty, we suspect the performance on "Sort by Property" instructions are lower because of how different optimal policies are for this task compared to others. This is the only instruction in which the goal state is defined by distance between particular items – a feature that may be harder to learn when there are multiple other objectives competing for representational space.
>
> > ii) For the HARD HOLDOUTS set-up, will the performance further improve if there are more easy combinations available for training? Are there any other possible solutions to address the easy-to-hard transfer problem?
>
> In Figure 6, we already vary the number of easier combinations used in training and find that by increasing those leads to improvement in performance. Note that we report only zero-shot performance on hard combinations, and expect that fine-tuning on some data from hard holdouts would further improve the performance.
>
> ------
>
> We thank you for the interesting questions! We hope our response has addressed the reviewer's concerns about the choice of environment.  If there are any further questions or concerns, we are happy to answer them here.

---

> > ### Comment · Reviewer_Pm6p · 2023-11-22
> >
> > Thanks for the authors' point-to-point and detailed responses. I appreciate the efforts they made to address my concerns. For the HARD HOLDOUTS set-up, I was actually expecting more results other than those in Figure 6. Given the author responses and comments from other reviewers, I will keep my rating as it is. However, I will not feel uncomfortable if other reviewers think the contribution is limited due to lack of novelty and/or insufficient experiment evaluation.

---

### Official Review · Reviewer_h2vj · 2023-11-02

**Soundness:** 3 good
**Presentation:** 3 good
**Contribution:** 2 fair
**Rating:** 5
**Confidence:** 5

**Summary:**

The paper presents a dataset and a system for compositional generalization to novel observation spaces, action spaces, and tasks using end-to-end modular architectures. These architectures divide the task into specialized differentiable modules for encoding observations and predicting actions; and they are connected. The authors create a controlled environment for testing. Experiments show that the modular approach allows agents to generalize to unseen combinations.

**Strengths:**

This paper presented an modular approach towards compositional generalization, which is an important field and is definitely relevant to the theme of the conference. The sections defining the model and describing the experiments are well-structured, effectively conveying the core concepts and findings of the paper.

**Weaknesses:**

The main weakness of the paper is its limited contribution.

On the dataset aspect, it's a bit unclear what's new in this dataset. So the proposed dataset is based on a 2D grid world, with some 3D rendering. I do not see significant differences between this setting and Minigrid (https://github.com/Farama-Foundation/Minigrid), and many works have been built on Minigrid. For example, Minigrid also contains different observation spaces (symbolic vs. image), and there are also works that use text descriptions too. It also contains instruction-level compositionality. Another example in robotics is CompoSuite: A Compositional Reinforcement Learning Benchmark https://arxiv.org/pdf/2207.04136.pdf

On the model aspect, I don't see significant differences between this paper and other papers that use (most times pretrained) LLMs for decision-making. For example, https://arxiv.org/pdf/2202.01771.pdf they have studied different encodings of the input too. Its also similar to GATO, except that GATO does not use "modules for encoding observations". Another example in robotics is Modular Lifelong Reinforcement Learning via Neural Composition https://openreview.net/pdf?id=5XmLzdslFNN

Findings. I think the findings of the paper are not completely new. Many aspects of it have been demonstrated in many works, mostly in more realistic settings such as robot manipulation. For example, GATO also demonstrated how it could learn new tasks faster based on other pretrained tasks. And again, the results here are only demonstrated in a very synthetic setting, it is very unclear how these findings can be generalized to realistic learning settings (e.g., multitask learning for robotics).

**Questions:**

N/A

---

> ### Author Response · Authors · 2023-11-21
>
> We are thankful to the reviewer for the positive comments, we are glad they found the topic relevant and the presentation clear! We however strongly disagree with the reviewer's claim about the lack of novelty in our work. As noted by the reviewer Pm6p, our work is novel in demonstrating that a modular architecture is an effective way to achieve compositional generalization wrt observation, action and instruction spaces; as well as for transfer from easy to hard combinations and to novel perceptual modalities. We also developed an environment that enables such study. In our response, we will clarify those contributions and key differences from prior work.
>
> ## Environment
>
> None of the existing environments, including the two the reviewer proposed (MiniGrid, CompoSuite), were suitable for studying the form of compositional generalization our work is primarily interested in, i.e. generalization to combinations of perception modalities, action and instruction spaces.
>
> The primary component lacking in Minigrid is a variety of action spaces – by limiting ourselves to only one action space, we would be missing a key factor of variation required for our study. Furthermore, MiniGrid supports one type of image observation (2D image), whereas our environment can generate an arbitrary number of visual observations by projecting the 3D space differently (see Figure 2). An important practical consideration is that MiniGrid codebase is _not constructed to support easy mixing and matching_ of the different types of observations, actions and instructions. With the extensive changes required to support the type of work we were interested in, construction of an entirely new codebase was more feasible than an extension of the existing MiniGrid codebase.
>
> CompoSuite supports easy combining of the following 4 elements: robotic arm, object id, obstacle id and instruction (with 4 options for each). However, different modalities of observation spaces are notably missing in CompoSuite. This is an important consideration when the aim is to train agents that can ingest different modalities (such as text, images, actuator positions) that warrant the use of specialized architectures for representation. Moreover, the four elements in CompoSuite do not map clearly to variations in observation, action and instruction distinction, which are the focus of our paper.
>
> In conclusion, the contributions in our environment are:
> 1. easy specification of each env instance as an explicit combination of perception, action and instruction space,
> 2. large number of perception modalities, action and instruction spaces,
> 3. easy addition of new spaces,
> 4. automated generation of expert data via A* solver.
>
> ## Model
>
> While, as the reviewer notes, each of the elements of our architecture is not unique, the specific combination is uniquely well suited for the settings we are interested in. The architectures used in GATO and LID, while also relying on transformer architectures, do not use composable modules to capture different observation and action spaces. The GATO architecture (also used in multi-modal multi-task settings), relies on tokenization and serialization of observation inputs, which will have variable length and would not allow easy incorporation of new observation modalities without affecting the performance on the pretraining tasks.
>
> The modules in (Mendez et al., 2022) specialize in _different subtasks_, instead of different observation and action modalities. Furthermore, in their work, the modules are searched over and composed in sequence to capture the subproblems in the overall task.
>
> ## Findings
>
> To the best of our knowledge, there is no other work that studies compositional generalization with respect to the three common and general factors of variation: observation, action and instruction spaces. While GATO does demonstrate transfer to new tasks, there are no experiments that would shed light on the compositional generalization wrt perception and action modalities. There are also no experiments demonstrating transfer specifically to new perception modalities.
>
> Furthermore, none of the experiments in GATO demonstrate that fine-tuning on the downstream tasks can be done without affecting performance on the remaining tasks, which is one of the demonstrated properties of our model. Finally, note that while large-scale experiments in realistic settings are often desirable, for the sake of drawing out clear empirical patterns (such as compositional generalization with respect to specific factors of variation), controlled experiments in synthetic domains are more useful.
>
> -----
>
> In conclusion, we argue that overall, the contribution and the insights gained are sufficiently different from prior work to warrant a significant contribution. These clarifications will be added to the updated version of the paper. We hope our response addressed your concerns, if there are any remaining concerns, we are happy to discuss them here.

---

### Official Review · Reviewer_n2i7 · 2023-11-06

**Soundness:** 3 good
**Presentation:** 4 excellent
**Contribution:** 2 fair
**Rating:** 5
**Confidence:** 3

**Summary:**

This paper proposes Compositional Interfaces (COIN) architectures, the end-to-end modular architectures for compositional generalization to unseen combinations of observation and action spaces in embodied agent settings.
Differentiable modules in COIN handle observation encoding and action prediction.
Each observation or action space has a module, but the controller is shared.
The environment with compositional structure is developed to investigate the architecture.
An environment instance is generated by combining observation, action, and instruction space.
The experiments show COIN enables compositional generalization and transfer learning. It generalizes to unseen combinations, and novel observation modalities can be quickly integrated.

**Strengths:**

- The problem of compositional generalization for embodied agent settings is important.

- The paper also developed a flexible compositional environment.

- It uses end-to-end training and does not require a special training scheme.

- The experiments support the ability of COIN for compositional generalization and transfer learning.

**Weaknesses:**

There are concerns that the task and the architecture do not cover general cases of compositional generalization in embodied agent tasks.

(1) **Disentangled inputs:**
It handles disentangled inputs since the observation, action, and instruction space are separately provided, but it does not address more general entangled inputs.

(2) **Types of combinations:**
There can be more possible compositional generalization problems, such as novel combinations of shape and color in observation.

(3) **Given component IDs:**
The IDs of observation and action spaces are provided for each sample to select modules. They can be hidden in more general cases.

Here are some other concerns.

(4) The agent is trained with imitation learning, while another widely used algorithm is reinforcement learning.

(5) In the experiment, the environment is small and synthetic. It also lacks strong baselines of existing embodied agent algorithms.

**Questions:**

Please refer to the weakness section.

---

> ### Author Response · Authors · 2023-11-20
>
> We thank the reviewer for the positive feedback!
> We'll address the reviewer's concern about the limited generality in the following.
>
> > (1) Disentangled inputs: It handles disentangled inputs since the observation, action, and instruction space are separately provided, but it does not address more general entangled inputs.
>
> > (3) Given component IDs: The IDs of observation and action spaces are provided for each sample to select modules. They can be hidden in more general cases.
>
> We do only consider the case where the instruction and observations are provided separately, and the observation and action IDs are known. However, we do not think this is a significant limitation of our work in terms of its impact and contribution. Instruction and observation are typically provided together (in fact, this is a default assumption in the [instruction following](https://arxiv.org/abs/1906.03926) literature and goal-conditioned sequential decision making). Similarly, the observation and action IDs are often known – the action space typically needs to be known in order to enact the action in the environment, while the observation space can be determined from the format of the input.
>
> > (2) Types of combinations: There can be more possible compositional generalization problems, such as novel combinations of shape and color in observation.
>
> In this work, we chose to specifically focus on compositional generalization with respect to combinations of observations, actions and instructions, as those are particularly relevant for generalist (multi-task, multi-modal) agents. As the reviewer suggests, there is lots of prior work on other kinds of compositional generalization -- that is good too!  We of course neither attempt nor claim to cover every possible instance of "compositionality"; similarly most other works on compositionality do not attempt all instances.  Even _discussing_ (much less algorithmically treating) all possible generalization problems is beyond the scope of a monograph, much less a single ICLR submission.
>
> > (4) The agent is trained with imitation learning, while another widely used algorithm is reinforcement learning.
>
> While adding comparison with other kinds of objectives would further strengthen our results, we argue that using only imitation learning is not a significant limitation of our work. Since the question of transfer and generalization is mostly related to the architecture and training regime, the results are also informative for the agents trained with reinforcement learning. The most related published works, such as [multi-game decision transformers](https://arxiv.org/abs/2205.15241) and [GATO](https://arxiv.org/abs/2205.06175), also train their agents with expert data only (either via offline RL or imitation learning).
>
> > (5) In the experiment, the environment is small and synthetic.
>
> While the larger scale and more realistic environments are often desirable, we argue that for the sake of drawing out clear and consistent empirical patterns, controlled environments such as the one developed in this paper are more useful. Such environment enabled us to: (a) generate a very large number of environment combinations and individual spaces, with which we can extract more signal from the noise, (b) have a much better control of the factors of variation – in our case, we can only vary the observation modality, action space and instruction space – without introducing other confounding sources of variation that would be present in more complex environments
>
> > It also lacks strong baselines of existing embodied agent algorithms.
>
> With respect to the choice of baselines, the literature has no standard off-the-shelf approaches applicable to our setting beyond our comparisons to single-task models.  The closest possible approaches are tokenized transformers such as used in GATO (which is not open-sourced).  In particular, "how to tokenize" would become a non-trivial question beyond the scope of this work, even if GATO was open-sourced.
>
> Hence the most fair comparison is with respect to agents trained in a single-task setting or to the performance in the expert data (in our case, the experts always complete the task). Note that for the same reasons, GATO also does not compare their model to alternative multi-task multi-modal architectures.
>
> -------
>
> We hope our response addressed your concerns, if there are any remaining concerns, we are happy to discuss them here!

---

### Comment · Area_Chair_KeEv · 2023-11-20
**Discussion between authors and reviewers**

Dear Reviewers,

Thanks for the reviews. The authors have uploaded their responses to your comments, please check if the rebuttal address your concerns and if you have further questions/comments to discuss with the authors. If the authors have addressed your concerns, please adjust your rating accordingly or vice versa.

AC

---

### Meta-Review · Area_Chair_KeEv · 2023-12-04

**Metareview:**

This paper investigates how to enable agents with compositional generalization to unseen combinations(CG) of observations, actions, and instructions. It shows that modular architectures exhibit such CG ability. The environment with compositional structure is developed to evaluate the architecture.  The experiments demonstrate the quick adaptation of the modules to the new observation.

Strengths:
+ The paper develops a flexible compositional environment for studying CG of agents.
+ The proposed modular approach towards compositional generalization supports end-to-end training.
+ Experiments show improvement over various COIN baselines.

Weaknesses:
- The proposed method can handle disentangled input but not the general entangled cases
- The typed of combinations are limited and do not cover novel combinations of shape and color in observation.
- Experiments do not compare with non-COIN baselines. It is hard to assess the effectiveness of this method.
- Limited novelty: the paper does not contain significant contributions in terms of model architectures, learning algorithms, and new findings.
- The environment is small and synthetic, similar as Minigrid and CompoSuite. It's not clear how to generalize to realistic settings with noise.

**Justification For Why Not Higher Score:**

Although the paper introduces a new environment/dataset for evaluating compositonal gerneralization ability of agents. Comparing to CompSuite, where the observation/action spaces naturally derive from robotics settings, it offers limited contribution to the community. Similarly, the paper does not contain significant contributions in terms of model architectures, learning algorithms, and new findings.
I recommend not to accept this paper at this stage, giving the authors more time to improve their paper.

**Justification For Why Not Lower Score:**

N/A

---

### Decision · Program_Chairs · 2024-01-16

Reject